# Signal-to-Noise Ratio Analysis for the Voltage-Mode Read-Out of Quartz Tuning Forks in QEPAS Applications

**DOI:** 10.3390/mi14030619

**Published:** 2023-03-08

**Authors:** Michele Di Gioia, Luigi Lombardi, Cristoforo Marzocca, Gianvito Matarrese, Giansergio Menduni, Pietro Patimisco, Vincenzo Spagnolo

**Affiliations:** 1PolySense Lab, Dipartimento Interateneo di Fisica, University and Politecnico of Bari, Via Amendola 173, 70126 Bari, Italy; 2Dipartimento di Ingegneria Elettrica e Dell’Informazione, Politecnico of Bari, Via Edoardo Orabona 4, 70126 Bari, Italy

**Keywords:** quartz-enhanced photoacoustic spectroscopy, quartz tuning fork, voltage-mode read-out, front-end electronics, signal-to-noise ratio, gas sensing

## Abstract

Quartz tuning forks (QTFs) are employed as sensitive elements for gas sensing applications implementing quartz-enhanced photoacoustic spectroscopy. Therefore, proper design of the QTF read-out electronics is required to optimize the signal-to-noise ratio (SNR), and in turn, the minimum detection limit of the gas concentration. In this work, we present a theoretical study of the SNR trend in a voltage-mode read-out of QTFs, mainly focusing on the effects of (i) the noise contributions of both the QTF-equivalent resistor and the input bias resistor R_L_ of the preamplifier, (ii) the operating frequency, and (iii) the bandwidth (BW) of the lock-in amplifier low-pass filter. A MATLAB model for the main noise contributions was retrieved and then validated by means of SPICE simulations. When the bandwidth of the lock-in filter is sufficiently narrow (BW = 0.5 Hz), the SNR values do not strongly depend on both the operating frequency and R_L_ values. On the other hand, when a wider low-pass filter bandwidth is employed (BW = 5 Hz), a sharp SNR peak close to the QTF parallel-resonant frequency is found for large values of R_L_ (R_L_ > 2 MΩ), whereas for small values of R_L_ (R_L_ < 2 MΩ), the SNR exhibits a peak around the QTF series-resonant frequency.

## 1. Introduction

Quartz-enhanced photoacoustic spectroscopy (QEPAS) is a well-known technique used for the detection of specific trace gases in complex mixtures [1,2,3,4,5,6,7]. The high performances in terms of selectivity and sensitivity allow for the exploitation of this technique in a wide range of applications, such as environmental monitoring [8,9,10], chemical analysis [11,12], and advanced biomedical diagnostics [13,14]. In QEPAS, acoustic waves are generated between the prongs of a quartz tuning fork (QTF) by the absorption of modulated light from the gas molecules, via non-radiative relaxation processes [2,15]. QTFs are employed as piezoelectric sensitive elements to transduce pressure waves in an electric signal [2,6]. This technique was firstly introduced in 2002 and exploited standard QTFs resonating at 32 kHz [16]. These quartz resonators are characterized by a good immunity to environmental acoustic noise because of their high quality factors (Q) and compact dimensions. Due to the sharp resonance, external noise sources outside of the resonator’s small bandwidth (~4 Hz at atmospheric pressure) do not influence the QTF signal [17]. 

Suitable front-end electronics must be designed to read-out the signal generated by the QTF. The common read-out architecture employed in QEPAS sensors is the transimpedance amplifier (TIA) [8,9,10,11,12,13,14], schematically depicted in Figure 1. 

In this configuration, the current generated by the QTF (I_qtf_) due to the charge displacement caused by piezoelectric effect when pressure waves put prongs in vibration is forced to flow entirely in the feedback resistor R_F_. Due to the virtual ground established on the inverting node of the operational amplifier (OPAMP), the measurement is insensitive to the stray capacitance C_st_ and, in turn, the output voltage (V_out_) is proportional to I_qtf_.

Recently, it has been demonstrated that a voltage-mode approach for the read-out electronics of QTFs can be advantageous in terms of signal-to-noise ratio (SNR) [18,19,20,21]. In this case, the piezoelectric transducer is coupled to a voltage amplifier, realized by means of a classic OPAMP in a non-inverting configuration, as illustrated in Figure 2, and characterized by very high input impedance. Since the QTF is an open circuit at DC, a load resistor R_L_ must necessarily be connected to the non-inverting input of the amplifier to establish its DC voltage to ground, when the input bias current of the OPAMP can be neglected.

The thermal noise of the resistor R_L_ contributes to the overall noise spectral density of the circuit output. Furthermore, the QTF experiences the loading effect due to R_L_; then, the input voltage V_qtf_ of the amplifier results from the product of the current generated by the QTF under this load and the value of R_L_ itself. As a consequence, R_L_ has a direct impact on both the level of the useful signal V_out_ and the noise at the output of the preamplifier. 

QEPAS requires synchronous detection techniques based on lock-in amplifiers (LIAs) to efficiently extract the useful signal component from the noise floor [2,22,23,24,25,26]. In LIAs, the amplifier signal is first multiplied with both a sinewave and a 90° phase-shifted copy of that sinewave at a selected operating frequency (f_op_); then, a low-pass filter (LPF) is used to retrieve the signal component at f_op_ or its multiples [22,26]. The phase noise in high-Q mechanical oscillators, such as QTFs, could be detrimental when LIAs are employed, since the amplitude of the output signal, and, in turn, the SNR, will be affected [27].

Both QTF signal and noise sources undergo the signal conditioning chain composed of the front-end voltage amplifier, the multiplication with a sine wave at frequency f_op_, and the low-pass filter. Therefore, a study on the SNR as a function of (i) the parameters of the amplifier components, (ii) the operating frequency, and (iii) the low-pass filter bandwidth must be carefully conducted to fully exploit the resonance properties of the QTF and maximize the performances of a QEPAS sensor.

In this work, we reported the effect of the R_L_ value on the frequency where the SNR exhibits its peak at the output of the lock-in amplifier, i.e., the optimum operating frequency for the QEPAS system. Starting from the well-known Butterworth–Van Dyke model for the QTF [17,28,29,30], we derived an analytical expression for the amplitude of the signal generated by the QTF as a function of frequency at different R_L_ values. For our purposes, only the main electronic noise contributions were considered, whereas phase noise was not taken into account. Furthermore, a mathematical modelling was developed with MATLAB for all the relevant contributions to the total noise spectral density at the output of the preamplifier, making possible a comparison among their relative weights. Finally, the behavior of the SNR as a function of the lock-in demodulation frequency was studied at different R_L_ values and LPF bandwidths, namely, the acquisition time. All the proposed analytical expressions were validated by comparing the developed MATLAB model with SPICE simulations, carried out considering a realistic model for the OPAMP used in the preamplifier. As a result of the study, general guidelines for the choice of the resistor R_L_ and the most suitable operating frequency for the QEPAS system implementing the voltage-mode read-out of QTFs can be derived. Furthermore, the proposed analysis allows for the study of the noise contributions at different bandwidths to optimize the acquisition time of QEPAS measurements.

## 2. Signal Response of a Quartz Tuning Fork Read Out in Voltage Mode

As mentioned above, the Butterworth–Van Dyke circuit and an equivalent Thevenin source were employed to model the QTF when excited by an acoustic wave [17] to find an analytical expression of the output signal V_out_ as a function of frequency for the circuit in Figure 2. For this investigation, we considered (i) an OPAMP with sufficiently high gain-bandwidth product so that the gain of the non-inverting configuration is A_v_ = V_out_/V_qtf_ = 1 + R_F_/R_S_, as shown in Figure 2; and (ii) the capacitance C_in_ between the non-inverting input and ground. Thus, the circuit to be studied is sketched in Figure 3. 

Here, the parasitic shunt capacitance C_p_ between the terminals of the QTF is in parallel with the input capacitance of the voltage amplifier C_in_, so that the total parasitic capacitance which loads the QTF is C_pt_ = C_p_ + C_in_. The voltage source V_in_ represents the internal electric signal generated by the piezoelectric effect, when the QTF is excited by an acoustic wave. Straightforward calculations provide the expression of the transfer function H_v_(jω) between the internal voltage V_in_(jω) and the output voltage V_out_(jω): (1)$$\begin{array}{c} H_{v}\left(j\omega \right)=\frac{V_{\mathrm{out}}\left(j\omega \right)}{V_{\mathrm{in}}\left(j\omega \right)}=\\ =A_{v}\frac{{j\omega C}_{s}R_{L}}{1-\omega^{2}\left({\mathrm{LC}}_{s}+R_{p}R_{L}C_{\mathrm{pt}}C_{s}\right)+j\omega \left[\left(R_{p}+R_{L}\right)C_{s}+R_{L}C_{\mathrm{pt}}-\omega^{2}{\mathrm{LC}}_{\mathrm{pt}}C_{s}R_{L}\right]}. \end{array}$$

The squared modulus of this transfer function, which describes how the squared amplitude of the circuit response depends on the frequency, can be written as follows:(2)$${\left|H_{v}\left(j\omega \right)\right|}^{2}=\frac{A_{v}^{2}}{\frac{{\left(1-\frac{\omega^{2}}{\omega_{R}^{2}}\right)}^{2}}{\omega^{2}R_{L}^{2}C_{s}^{2}}+{\left(1+\frac{C_{\mathrm{pt}}}{C_{s}}\right)}^{2}{\left[\left(1-\frac{\omega^{2}}{\omega_{P}^{2}}\right)+\frac{R_{P}}{R_{L}}\frac{C_{s}}{C_{\mathrm{pt}}+C_{s}}\right]}^{2}},$$
where $\omega_{R}=\frac{1}{\sqrt{{\mathrm{LC}}_{s}+R_{p}R_{L}C_{\mathrm{pt}}C_{s}}}$ is the angular frequency in which the transfer function H_v_(jω) exhibits a real value and $\omega_{P}=\frac{1}{\sqrt{L\frac{C_{s}C_{\mathrm{pt}}}{C_{s}+C_{\mathrm{pt}}}}}=\frac{1}{\sqrt{{\mathrm{LC}}_{\mathrm{eq}}}}$ is the parallel-resonant angular frequency of the QTF as loaded in the circuit shown in Figure 3.

AC SPICE simulations have been carried out to confirm the validity of the expression in Equation (2). The set of typical parameters listed in Table 1 has been used for the QTF in both the analytical model and SPICE simulations. Moreover, in SPICE simulations, C_in_ is included in the model of the AD8067, provided by Analog Devices. AD8067 is a low-noise, high speed, FET input operational amplifier. Thanks to its very low input bias current, it is suitable for precision and high gain applications [31].

The series-resonant frequency $f_{S}\;$of the QTF is:$$f_{S}=\frac{\omega_{S}}{2\pi }=\frac{1}{2\pi \sqrt{{\mathrm{LC}}_{s}}}=32768\;\mathrm{Hz},$$
whereas its quality factor Q is
$$Q=\frac{1}{\omega_{s}R_{p}C_{s}}=10^{4},$$
which are typical values for a standard QTF used in QEPAS sensors [2,16,17,26]. Moreover, the gain of the non-inverting configuration is A_v_ = 48 and the parallel-resonant frequency of the QTF is f_P_ = ω_P_/2π = 32,781 Hz. The value of C_P_ ≅ C_P_ + C_S_ could be found by measuring the equivalent capacitance of the QTF at low frequencies using the capacitance-voltage profiling technique. The ratio C_S_/C_P_ is extracted by the ratio f_P_/f_S_ after measuring the parallel and series-resonant frequencies of the QTF, where the sensor exhibits the minimum and maximum admittance, respectively. Last, L can be found from f_S_, knowing C_S_, and R_P_ is extracted from the quality factor Q of the QTF [21]. Figure 4 shows the comparison between the results provided by the SPICE simulations and the analytical model for three different values of R_L_ (100 kΩ, 0.5 MΩ, and 2.5 MΩ).

The perfect matching between two modellings demonstrates that Equation (2) can be used to accurately represent the behavior of the circuit in Figure 3. The maximum difference between the peak frequencies of corresponding curves is in the order of a few hundredths of Hz and the mean absolute percentage error between corresponding curves is about 1.8%. 

### 2.1. Peak Frequency of the QTF Signal Response as a Function of R_L_

As shown in Figure 4, the output signal amplitude and the peak frequency f_peak_ strongly depend on the value of the resistor R_L_. This is particularly relevant for an optimal choice of the operating frequency in the QEPAS technique, aimed at exploiting as much as possible the resonance properties of the QTF.

Figure 5 shows the f_peak_ trend as a function of R_L_. This curve was retrieved computing |H_v_(jω)|^2^ for different values of R_L_ and then applying MATLAB “max” function to yield f_peak_ values.

For R_L_ values lower than 100 kΩ, f_peak_ tends to the series-resonant frequency f_S_ = 32768 Hz; whereas, for values of R_L_ higher than 2 MΩ, f_peak_ tends to assume the values of the parallel-resonant frequency f_P_, in our case equal to 32781 Hz.

The behavior of the peak position as a function of the value of R_L_ can be explained considering the two terms which compose the denominator of the function |H_v_(jω)|^2^ in Equation (2), reported here, below, for ease of reading, as Den1 and Den2:(3)$$\mathrm{Den}1\left(\omega \right)=\frac{{\left(1-\frac{\omega^{2}}{\omega_{R}^{2}}\right)}^{2}}{\omega^{2}R_{L}^{2}C_{s}^{2}},\;\mathrm{Den}2\left(\omega \right)={\left(1+\frac{C_{\mathrm{pt}}}{C_{s}}\right)}^{2}{\left[\left(1-\frac{\omega^{2}}{\omega_{P}^{2}}\right)+\frac{R_{P}}{R_{L}}\frac{C_{s}}{C_{\mathrm{pt}}+C_{s}}\right]}^{2}$$

The behavior of Den1 and Den2 as a function of frequency is reported in Figure 6a,b for R_L_ = 100 kΩ and R_L_ = 10 MΩ, respectively.

Results show that Den1(ω) is strongly dependent on the value of R_L_. In the investigated frequency range: (i) the maximum value of Den1 varies from 868.3 in Figure 6a to 0.17 in Figure 6b; (ii) the blue curve at 100 kΩ varies more rapidly than the Den1 curve at 10 MΩ in the frequency range close to the minimum; and (iii) the frequency where the Den1 minimum occurs decreases by more than 20 Hz, from 32,768 Hz in Figure 6a to 32,746 Hz in Figure 6b. Conversely, the dependence of Den2(ω) on R_L_ is only due to the term $\frac{R_{P}}{R_{L}}\frac{C_{s}}{C_{\mathrm{pt}}+C_{s}}$, which is negligible when R_L_ ≥ 1 MΩ, since C_s_ << C_pt_ and R_P_ << R_L_. Indeed: (i) the red curve maximum value varies from 30.8 in Figure 6a to 21.5 in Figure 6b; (ii) Den2 value variations in the frequency range close to the minimum value are slightly different at 100 kΩ and at 10 MΩ; and (iii) the difference frequency where the Den2 minimum occurs is about 13 Hz, from 32,794 Hz in Figure 6a to 32,781 Hz in Figure 6b. As a result, for R_L_ < 100 kΩ, the contribution of Den1(ω) becomes dominant, so that when R_L_ decreases, the minimum value of the denominator of |H_v_(jω)|^2^ tends to the zero of Den1(ω) function, located at ω = ω_R_. Moreover, in this range of R_L_ values, ω_R_ could be approximated to ω_S_: $$\omega_{R}=\frac{1}{\sqrt{{\mathrm{LC}}_{s}+R_{p}R_{L}C_{\mathrm{pt}}C_{s}}}≅\frac{1}{\sqrt{{\mathrm{LC}}_{s}}}=\omega_{S},$$
and the peak of |H_v_(jω)|^2^ tends to the series-resonant frequency of the QTF. Instead, for R_L_ > 2 MΩ, Den1(ω) becomes less relevant and Den2(ω) tends to be dominant in the sum of the two terms. As a consequence, the minimum of the sum tends to the zero of Den2(ω). Finally, in this range of R_L_, this zero is very close to ω = ω_P_, thus, the peak of |H_v_(jω)|^2^ is almost coincident with the parallel-resonant frequency of the QTF.

### 2.2. Peak of the QTF Signal Response as a Function of R_L_

The trend of the peak value of |H_v_(jω)|^2^ as a function of R_L_ value is shown in Figure 7. The same method applied for Figure 5 was used to obtain this curve.

The peak value of |H_v_(jω)|^2^ is an increasing function of R_L_ up to a saturation value at R_L_ > 1 GΩ. As discussed in the previous section, for large values of R_L_, ω_peak_ = 2πf_peak_ is very close to the parallel-resonant angular frequency ω_p_ and the peak of |H_v_(jω)|^2^ tends to the following value: (4)$${\left|H_{v}\left({j\omega }_{\mathrm{peak}}\right)\right|}^{2}≅{\left|H_{v}\left({j\omega }_{P}\right)\right|}^{2}≅A_{v}^{2}\frac{\omega_{P}^{2}R_{L}^{2}C_{s}^{2}}{{\left(1-\frac{\omega_{P}^{2}}{\omega_{R}^{2}}\right)}^{2}}$$

The denominator of Equation (4) can be rewritten as:$$1-\frac{\omega_{P}^{2}}{\omega_{R}^{2}}=1-\frac{{\mathrm{LC}}_{s}+R_{p}R_{L}C_{\mathrm{pt}}C_{s}}{{\mathrm{LC}}_{\mathrm{eq}}}=\frac{L\left(C_{\mathrm{eq}}-C_{s}\right)-R_{p}R_{L}C_{\mathrm{pt}}C_{s}}{{\mathrm{LC}}_{\mathrm{eq}}}≅-\frac{R_{p}R_{L}C_{\mathrm{pt}}}{L}$$

Since C_Equation_ ≅ C_s_ and R_L_ is very large, it results
$${\left|H_{v}\left({j\omega }_{\mathrm{peak}}\right)\right|}^{2}≅A_{v}^{2}\frac{\frac{1}{{\mathrm{LC}}_{\mathrm{eq}}}R_{L}^{2}C_{s}^{2}}{\frac{R_{p}^{2}R_{L}^{2}C_{\mathrm{pt}}^{2}}{L^{2}}}≅A_{v}^{2}\frac{{\mathrm{LC}}_{s}}{R_{p}^{2}C_{\mathrm{pt}}^{2}},$$
which is independent on R_L_.

However, the performance of the QEPAS technique will depend on the SNR obtained at the output of the preamplifier, not only on the amplitude of the signal. Therefore, it is mandatory to carry out a detailed study of the electronic noise contributions that are involved in the circuit, with the purpose of finding out the optimal operating frequency maximizing the SNR.

## 3. Contributions to the Output Noise Spectral Density

The most relevant contributions to the total electronic noise at the output of the voltage-mode preamplifier are shown in Figure 8. 

In our calculation, the phase noise was neglected [27] and only the main electronic noise contributions were considered. Each resistor R_i_ of the circuit has been associated to its thermal noise voltage source, e_n_i_^2^ = 4kTR_i_. The OPAMP noise has been characterized by means of the classic equivalent input noise voltage (e_n_op_) and current (i_n+_ and i_n−_) sources. To simplify the study without losing accuracy, it is possible to neglect the noise contributions of R_F_ and R_S_, composing the feedback network, due to the small values of these resistors. For the same reasons, the equivalent noise current i_n−_ associated to the inverting input of the OPAMP likewise does not give any relevant contribution. Moreover, all the sources in Figure 8 can be considered independent, so that the total output noise power spectral density S_ntot_(ω) can be evaluated as follows:(5)$$S_{\mathrm{ntot}}\left(\omega \right)≅S_{\mathrm{np}}\left(\omega \right)+S_{\mathrm{nL}}\left(\omega \right)+S_{\mathrm{nop}}\left(\omega \right)+S_{\mathrm{nin}+}\left(\omega \right),$$
where the terms of the right-hand side come from R_p_, R_L_, e_n_op_, and i_n+_, respectively. In Equation (5), the single terms are the product of the spectral density of each noise source multiplied by the squared modulus of the transfer function between the noise source and the output of the circuit, determined using the superposition principle [32]. 

Concerning the noise associated to the resistor R_p_, the transfer function $S_{\mathrm{np}}\left(\omega \right)$ between the source e_n_p_ and the output of the preamplifier is H_v_(jω), thus:$$S_{\mathrm{np}}\left(\omega \right)=4{\mathrm{kTR}}_{p}{\left|H_{v}\left(j\omega \right)\right|}^{2}.$$

As a consequence, the behavior of S_np_(ω) as a function of both the frequency and R_L_ is the same discussed in Section 2.

Let us now consider the noise contribution from the resistor R_L_. The transfer function between the source e_n_l_ and the output of the front-end is:(6)$$\begin{array}{cc} H_{L}\left(j\omega \right) & =\frac{V_{n\_\mathrm{out}}\left(j\omega \right)}{e_{n\_l}\left(j\omega \right)}\\ & =A_{v}\frac{1-\omega^{2}{\mathrm{LC}}_{s}+{j\omega R}_{p}C_{s}}{1-\frac{\omega^{2}}{\omega_{R}^{2}}+\;j\omega \left[\left(R_{L}+R_{p}\right)C_{s}+{\;R}_{L}C_{\mathrm{pt}}-{\;\omega }^{2}{\mathrm{LC}}_{\mathrm{pt}}C_{s}R_{L}\right]}. \end{array}$$

The denominators of H_v_(jω) and H_L_(jω) are the same and the two transfer functions differ only in their numerator. The contribution of the thermal noise of the resistor R_L_ to the total output noise spectral density is:(7)$$S_{\mathrm{nL}}\left(\omega \right)=4{\mathrm{kTR}}_{L}{\left|H_{L}\left(j\omega \right)\right|}^{2}.$$

Figure 9 shows the behavior of S_nL_ as a function of the frequency, for four different values of R_L_ (100 kΩ, 500 kΩ, 6 MΩ, and 20 MΩ). 

The curves calculated with Equation (7) and those obtained by SPICE simulation are in excellent agreement. The function |H_L_(jω)| has a minimum located at ω = ω_S_, since, at the series-resonant frequency, the Butterworth–Van Dyke impedance model of the QTF is reduced to R_p_, which is its minimum value. Accordingly, also S_nL_(ω) exhibits a minimum at the same frequency. Furthermore, a peak appears around the parallel-resonant frequency for increasing values of R_L_. Figure 10 reports the behavior of the peak value of S_nL_(ω) as a function of R_L_, which varies from 500 kΩ to 20 MΩ.

Using the set of parameters listed in Table 1, starting from low values of R_L_, this peak appears when R_L_ ≅ 500 kΩ (dashed magenta and green curves in Figure 9), then increases up to R_L_ ≅ 6 MΩ (Figure 10). Beyond this value, the peak decreases for increasing values of R_L_ and the function S_nL_(ω) assumes lower values in the frequency range under investigation.

Figure 7 and Figure 10 suggest that, at least for what concerns the noise contributions considered up to now, it is convenient to work with large values of the resistor R_L_ because the amplitude of the output signal increases with R_L_ (see Figure 7) and the noise contribution due to this resistor decreases (see Figure 10), whereas the contribution from the thermal noise of R_p_ has exactly the same frequency behavior of the signal.

Let us now consider the contribution of the input equivalent voltage noise of the OPAMP to the total output noise: (8)$$S_{\mathrm{nop}}\left(\omega \right)=e_{n\_\mathrm{op}}^{2}{\left|H_{\mathrm{en}}\left(j\omega \right)\right|}^{2}.$$

Since the flicker noise has been considered negligible in the narrow bandwidth of interest, namely, around the QTF resonance frequency, the input equivalent noise voltage has a constant power spectral density, i.e., it can be considered as white noise. The value of e_n_op_ has been set at 6.6 nV/√Hz, as reported in the data sheet of the AD8067 [31]. The transfer function H_en_(jω) is the following:(9)$$\begin{array}{c} H_{\mathrm{en}}\left(j\omega \right)=\frac{V_{n\_\mathrm{out}}\left(j\omega \right)}{e_{n\_\mathrm{op}}\left(j\omega \right)}\\ =A_{v}\frac{1-\omega^{2}\left({\mathrm{LC}}_{s}+R_{p}R_{L}C_{p}C_{s}\right)+j\omega \left[\left(R_{p}+R_{L}\right)C_{s}+R_{L}C_{p}-\omega^{2}{\mathrm{LC}}_{p}C_{s}R_{L}\right]}{1-\omega^{2}\left({\mathrm{LC}}_{s}+R_{p}R_{L}C_{\mathrm{pt}}C_{s}\right)+\;j\omega \left[\left(R_{p}+R_{L}\right)C_{s}+{\;R}_{L}C_{\mathrm{pt}}-{\;\omega }^{2}{\mathrm{LC}}_{\mathrm{pt}}C_{s}R_{L}\right]}, \end{array}$$
in which the numerator differs from the denominator only for the capacitance C_pt_ = C_p_ + C_in_ replaced by C_p_. This contribution can be considered as constant in the frequency range investigated for QEPAS applications.

The contribution of the input equivalent current noise i_n+_ to the overall output noise spectral density S_ntot_(ω) is
(10)$$S_{\mathrm{nin}+}\left(\omega \right)=i_{n+}^{2}\left(\omega \right)R_{L}^{2}{\left|H_{L}\left(j\omega \right)\right|}^{2}.$$

Using Equations (10) and (7), the contributions to S_ntot_(ω), respectively due to i_n+_ and R_L_, can be compared: $$\frac{S_{\mathrm{nin}+}\left(\omega \right)}{S_{\mathrm{nL}}\left(\omega \right)}=\frac{i_{n+}^{2}\left(\omega \right)R_{L}}{4\mathrm{kT}}.$$

The contribution of S_in+_(ω) can be neglected with respect to S_nL_(ω) if
$$i_{n+}^{2}\left(\omega \right)\ll \frac{4\mathrm{kT}}{R_{L}}.$$

If we consider a large R_L_ value, i.e., 100 MΩ, S_in+_(ω) will be negligible at room temperature if i_n+_ << 13 fA/√Hz. Since the AD8067 has FET inputs, i_n+_(ω) cannot be considered white, but is a linear function of the frequency, in the range where the flicker noise can be neglected [33]. In our model, the value of i_n+_(ω) at 10 kHz has been set to 1 fA/√Hz, a slightly higher value than the one reported in the datasheet of the OPAMP, which is 0.6 fA/√Hz [31], and the slope of the linear function has been set to +20 dB/dec [33]. Figure 11 shows the excellent correspondence between the analytical model used for i_n+_(ω) and the input equivalent noise current resulting from a simulation carried out with the SPICE model of the AD8067. 

It is worth noticing that around the resonance frequencies of the QTF, the level of i_n+_(ω) remains lower than the limit of 13 fA/√Hz determined above. Thus, we can conclude that the contribution S_in+_(ω) can be neglected with respect to S_nL_(ω) in the noise analysis of our circuit without losing accuracy.

The analysis carried out so far allows for a comparison of the terms in Equation (5) to understand which ones are dominant for the output noise spectral density of the preamplifier. Figure 12 compares the spectral contributions of the OPAMP and the resistors R_p_ and R_L_ to the overall S_ntot_(ω) at the output of the circuit. The terms due to e_n_op_ and i_n+_, namely, S_nop_(ω) and S_nin+_(ω), respectively, have been summed, resulting in the overall noise contribution of the OPAMP (S_opamp_(ω)) and allowing for a comparison of the analytical model with the results of SPICE simulations, in which the two terms are not distinguishable. The same R_L_ values of Figure 9 have been considered (100 kΩ, 500 kΩ, 6 MΩ, and 20 MΩ). 

For all the considered cases, the results provided by the analytical model and the SPICE simulations exhibit a very good agreement. In addition, the contribution of the OPAMP to the total output noise is always negligible compared to the sum of the contributions from R_L_ and R_p_. S_nL_(ω) tends to prevail over S_np_(ω) for R_L_ < 6 MΩ (Figure 12a,b), whereas the opposite happens when the value of R_L_ is larger than 6 MΩ (Figure 12c,d). This confirms the previous conclusion about the advantage of working with large values of the resistor R_L_ in order to achieve good performance in terms of SNR.

## 4. Signal-to-Noise Ratio at the Output of the Lock-in Amplifier

Synchronous detection techniques based on laser modulation and lock-in amplifier are always used in QEPAS sensors to increase the SNR of the measurements. In the previous sections, the amplitude of the useful signal and the noise spectral density at the output of the voltage-mode preamplifier are expressed as a function of frequency. These expressions can be conveniently exploited to evaluate the SNR at the output of the LIA. Thanks to synchronous demodulation and narrow-band low-pass filtering, the LIA output signal is DC-level proportional to the amplitude of the preamplifier response at the operating frequency. When the preamplifier response is acquired at the QTF resonance frequency, the trend of the SNR can be investigated in a small range around f_s_. Thus, if the preamplifier response is acquired at a certain frequency f_op_ close to f_s_, the output signal is proportional to |H_v_(f_op_)|. For what concerns noise, as a result of demodulation, the LIA transfers around DC the noise spectrum at the output of the preamplifier, centered around the LIA reference frequency. This noise is then filtered by means of the LIA narrow low-pass filter. Thus, the LIA behaves in practice like a narrow band-pass filter centered around its reference frequency. Hence, LIA was modelled as a biquadratic band-pass filter with a transfer function:(11)$$H_{\mathrm{LIA}}\left(j\omega \right)=\frac{j\frac{\omega_{\mathrm{op}}}{Q_{\mathrm{filt}}}\omega }{\omega_{\mathrm{op}}^{2}-\omega^{2}+j\frac{\omega_{\mathrm{op}}}{Q_{\mathrm{filt}}}\omega },$$
characterized by unity gain at center frequency and −3dB bandwidth BW = ω_op_/Q_filt_ [34]. 

Therefore, we can describe the behavior of the SNR at the output of the LIA as a function of the chosen operating frequency f_op_ = ω_op_/2π by means of the following function: (12)$$\begin{array}{cc} {\mathrm{SNR}}_{n}\left(\omega_{\mathrm{op}}\right) & =\frac{\left|H_{v}\left(\omega_{\mathrm{op}}\right)\right|}{\sqrt{\int_{-\infty }^{\infty }S_{\mathrm{ntot}}\left(\omega \right){\left|H_{\mathrm{LIA}}\left(j\omega \right)\right|}^{2}d\omega }}=\\ & ≅\frac{\left|H_{v}\left(\omega_{\mathrm{op}}\right)\right|}{\sqrt{\int_{-\infty }^{\infty }S_{\mathrm{np}}\left(\omega \right){\left|H_{\mathrm{LIA}}\left(j\omega \right)\right|}^{2}\mathrm{df}+\int_{-\infty }^{\infty }S_{\mathrm{nL}}\left(\omega \right){\left|H_{\mathrm{LIA}}\left(j\omega \right)\right|}^{2}d\omega }} \end{array}$$
where the amplitude of the input signal V_in_ (see Figure 3) has been normalized to 1 V. 

By separating the noise contributions from R_p_ and R_L_, we can define two functions: $${\mathrm{SNR}}_{\mathrm{np}}^{2}\left(\omega_{\mathrm{op}}\right)=\frac{{\left|H_{v}\left(\omega_{\mathrm{op}}\right)\right|}^{2}}{\int_{-\infty }^{\infty }S_{\mathrm{np}}\left(\omega \right){\left|H_{\mathrm{LIA}}\left(j\omega \right)\right|}^{2}d\omega }$$
and
$${\mathrm{SNR}}_{\mathrm{nL}}^{2}\left(\omega_{\mathrm{op}}\right)=\frac{{\left|H_{v}\left(\omega_{\mathrm{op}}\right)\right|}^{2}}{\int_{-\infty }^{\infty }S_{\mathrm{nL}}\left(\omega \right){\left|H_{\mathrm{LIA}}\left(j\omega \right)\right|}^{2}d\omega }$$

Thus, the total normalized squared-SNR can be rewritten as follows:(13)$${\mathrm{SNR}}_{n}^{2}=\frac{1}{\frac{1}{{\mathrm{SNR}}_{\mathrm{np}}^{2}}+\frac{1}{{\mathrm{SNR}}_{\mathrm{nL}}^{2}}}.$$

First, let us consider a very narrow-bandwidth BW = 0.5 Hz, corresponding to a settling time of about 1.3 s for an equivalent first-order LIA low-pass filter. In this case, the behavior of the integrated noise as a function of f_op_ is expected to be very similar to the output noise spectral density. Thus, the SNR_np_ contribution is expected to be almost spectrally flat since the noise from R_P_ and the signal have the same transfer function H_v_(f). Nonetheless, considering a representative value for R_L_ = 100 kΩ, a peak of SNR_np_ emerges around the peak frequency of |H_v_(f)|, because of the effect of integration, as reported in Figure 13. As shown in Figure 13, for a bandpass filter, the effect of the integration bandwidth on the input noise spectral density can be more intuitively represented by means of a simple brick-wall filter. 

Indeed, in the narrow integration bandwidth, the function S_np_(f) takes monotonically decreasing values on both sides of f_op_ = f_peak_; whereas, for any other frequency value, S_np_(f) has decreasing values in one direction, but increasing values in the other, as illustrated in Figure 13. As a consequence, the value of the integrated noise normalized to the value of the signal reaches a minimum value at f_op_ = f_peak_, generating a peak value in the function SNR_np_(f_op_). 

The presence of a peak value for SNR_np_ is also visible in Figure 14, which reports SNR_np_ as a function of the frequency f_op_, for different values of R_L_. Moreover, for increasing values of R_L_, the peak moves from f_S_ to f_P_; for large values of R_L_, it becomes slightly more pronounced, even though the overall behavior always remains rather flat.

As for the term SNR_nL_, its behavior as a function of f_op_ is strongly affected by the minimum of the noise spectral density S_nL_ at f_S_. This effect prevails over the peak value of the signal, which moves towards f_P_ for increasing values of R_L_, resulting in a local maximum point on SNR_nL_ which is always located at the series-resonant frequency f_S_ for any value of R_L_. Figure 15 shows the behavior of SNR_nL_ as a function of the operating frequency for different R_L_ values. 

Higher values of SNR_nL_ are obtained for increasing values of R_L_ values. Therefore, as reported in Equation (13), SNR_np_ becomes dominant in the calculation of the overall SNR for large values of R_L_.

We can now consider the overall SNR at the LIA output given by Equation (13) as a function of f_op_. Figure 16 reports the SNR_n_ as a function of f_op_ for different values of R_L_ with BW = 0.5 Hz. Here, a comparison between the results given by the analytical model and the corresponding SPICE simulation is also proposed, highlighting an excellent correspondence. 

For low values of R_L_, SNR_np_ is almost flat (see Figure 14) and the peak of the total SNR_n_ coincides with the peak of SNR_nL_, so the best operating frequency for QEPAS is the series-resonant frequency f_S_ of the QTF. When R_L_ is increased, as already pointed out, the contribution of SNR_nL_ becomes less relevant, so a local peak in the SNR_n_ emerges, corresponding to the peak of SNR_np_, which occurs at the parallel-resonant frequency f_P_.

From Figure 16, it is evident that the SNR_n_ peak feature at f_S_ becomes less sharp as R_L_ increases. For R_L_ = 100 MΩ, the SNR_n_ becomes quite flat around f_S_ and a sharp peak appears at f_P_. In general, slightly better results in terms of SNR are obtained with large values of R_L_.

The study above has been carried out for a LIA with a very narrow-band filter, which is useful when the requirements in terms of noise rejection are very harsh and the long time needed for a single measurement can be tolerated. With fast measurements [35,36], the bandwidth of the LIA filter must necessarily be increased, and the results of the previous analysis must be revised. Moreover, the QTF response time represents a further issue to deal with when fast measurements are needed. The response time is given by:$$\tau =\frac{Q}{{\pi f}_{s}}≅100\;\mathrm{ms}.$$

This implies that, in conventional QEPAS applications, a long integration time (300 to 400 ms) is required to acquire the LIA output signal. Nevertheless, specific QEPAS techniques such as Beat Frequency (BF) QEPAS exploit the fast transient response of an acoustically excited QTF to retrieve the gas concentration, the resonance frequency, and the quality factor of the QTF [7]. This approach overcomes the limitations imposed by the time response of the QTF, allowing shorter acquisition times and faster measurements. 

As concerns the contribution to the total SNR at the LIA output due to the resistor R_p_, since the integration bandwidth is larger, the effect of the integration around f_peak_, described above (see Figure 13), is more relevant. Consequently, the peak of SNR_np_ as a function of the operating frequency, always placed at f_peak_, becomes sharper and more pronounced. Figure 17 shows the behavior of SNR_np_ as a function of f_op_ when BW = 5 Hz, corresponding to a settling time of about 130 ms for an equivalent first order LIA low-pass filter for the same values of R_L_ considered in Figure 14. 

We now analyze the R_L_ noise contribution to the SNR for increasing values of the LIA bandwidth. The effect of the minimum of S_nL_, placed at f_S_, tends to be less relevant, because of the increase of the integration bandwidth. Therefore, the function SNR_nL_ becomes flatter around the series-resonant frequency. Nonetheless, for small values of R_L_, the peak of the signal is located at f_S_ and, consequently, the peak of SNR_nL_ is still placed at f_S_. When R_L_ is increased, the peak of the signal moves towards the parallel-resonant frequency f_P_ and the effect of the minimum of S_nL_ becomes less relevant, so SNR_nL_ tends to be flat. For large values of R_L_, S_nL_ values decrease (see Figure 12c,d), whereas the peak of the signal placed at f_P_ increases; as a result, SNR_nL_ exhibits a peak at f_P_, which becomes sharper as R_L_ increases. The behavior of SNR_nL_ as a function of the operating frequency f_op_ is shown in Figure 18 for BW = 5 Hz.

Finally, from Equation (13), Figure 19 shows that the peak value of SNR_n_ results at f_S_, where the peak of both SNR_np_ and SNR_nL_ is located, for low values of R_L_. 

For increasing values of R_L_, the SNR_n_ peak feature starts to flatten out until, for values of R_L_ larger than about 2 MΩ, a peak emerges close to f_P_, where both SNR_np_ and SNR_nL_ have their maximum value. This peak becomes as sharper as R_L_ increases, as shown in Figure 19, which represents the behavior of SNR as a function of f_op_ for BW = 5 Hz. Additionally in this case, SPICE simulations are in very good agreement with the results provided by the analytical model. The SNR_n_ peak at 100 MΩ is more than two times higher than the values close to f_s_.

Finally, the presented study can be applied to compute the Normalized Noise Equivalent Absorption (NNEA), an important parameter to compare QEPAS sensors [2,4,5,7,10]. NNEA is defined as follows:$$\mathrm{NNEA}=\frac{P\cdot \alpha }{\mathrm{SNR}\cdot \sqrt{\Delta f}}$$
where P is the laser optical power, α is the absorption coefficient of the gas under investigation, SNR is the signal-to-noise ratio, and Δf is the integration bandwidth.

In [10], an NNEA of 5.0 × 10^−9^ W∙cm^−1^/√Hz for the detection of CO_2_ in an N_2_ mixture, employing a transimpedance amplifier with a 10-MΩ feedback resistor and a narrow-bandwidth LIA filter, was demonstrated. Assuming the same value of NNEA for a voltage preamplifier with a 10-MΩ bias resistor and an integration bandwidth of 0.5 Hz, a 100-MΩ bias resistor and the same integration bandwidth would lead to an NNEA of 4.4 × 10^−9^ W·cm^−1^/√Hz, thus an improvement of a factor of 1.1, as suggested from Figure 16.

Considering a 5 Hz LIA filter bandwidth, an NNEA of 5.8 × 10^−9^ W∙cm^−1^/√Hz can be calculated for a bias resistor of 10 MΩ. An improvement of a 1.4 factor can be calculated when a bias resistor of 100 MΩ is employed, leading to a NNEA of 4.1 × 10^−9^ W·cm^−1^/√Hz. Table 2 summarizes the NNEA values obtained for different values of R_L_ and Δf: it can be observed that increasing R_L_ at a fixed filter bandwidth always results in an improvement of the NNEA.

## 5. Discussion and Conclusions

In this work, we investigated the SNR trend as a function of the operating frequency for a voltage-mode read-out of QTFs, comparing the results obtained by the developed model with SPICE simulations. The effects of the R_L_ values, the operating frequency, and the bandwidth of the LIA low-pass filter on the overall SNR were investigated. We firstly demonstrated that the contribution of the OPAMP to the output noise of the voltage preamplifier can be always neglected and the output noise power density is dominated by the contributions of the QTF-equivalent resistor R_p_ and R_L_. Moreover, the dominant contribution to the total noise changes from R_L_, when R_L_ < 10 MΩ, to R_p_, for R_L_ > 10 MΩ.

According to our model, the peak value of the SNR tends to increase along with the bias resistor R_L_. 

When a narrowband low-pass filter of the LIA is employed (e.g., bandwidth of 0.5 Hz), the curves shown in Figure 16 suggest that only a limited increase of the SNR can be obtained by either increasing the value of the resistor R_L_ or varying the operating frequency. For R_L_ < 10 MΩ, the best operating frequency for the QEPAS technique is the series-resonant frequency of the QTF f_S_, where SNR function exhibits a peak value. This peak becomes less pronounced when R_L_ is increased. A 10-MΩ bias resistor ensures a 1.3 times higher SNR at the series-resonant frequency f_S_, with respect to a 100 kΩ-bias resistor. For large values of R_L_, SNR tends to be flat around f_S_ and a small peak emerges at the parallel-resonant frequency f_P_. For instance, increasing the value of R_L_ up to 100 MΩ leads to a further increase of the SNR peak by a factor of 1.4. Thus, for these large values of R_L_, the choice of the optimal operating frequency for QEPAS is not very critical, in case of a narrow-band LIA filter. 

When a wider LPF bandwidth of the LIA is employed (e.g., bandwidth of 5 Hz), the SNR peak as a function of the operating frequency is still placed at f_S_ for small values of R_L_, as in the case of narrow-band filter (see Figure 19). For large values of R_L_, the SNR peak at f_P_ tends to be sharper as compared to the case of narrow-band LIA filter, thus the operating frequency for the QEPAS technique must be chosen as close as possible to f_P_ to maximize SNR. As an example, for R_L_ = 100 MΩ, the SNR peak is 1.4 times higher with respect to R_L_ =10 MΩ.

In addition, the parallel-resonant frequency f_P_ of the QTF depends on the input capacitance of the OPAMP, so it is not an intrinsic property of the sensitive element. Thus, suitable techniques must be used to measure f_P_ in presence of C_in_ to exploit large values of R_L_, increase SNR, and optimize the performance of the QEPAS sensor, especially when short acquisition times are needed. Instead, for long acquisition times, large values of R_L_ are not very effective for the increase of the overall SNR, and an accurate setting of the operating frequency close to f_s_ is not critical because of the flatness of SNR as a function of f_op_. Of course, the value of R_L_ cannot be made too large, since the input bias current of the OPAMP would cause unacceptable input offset levels. All the results obtained with the previous analysis have been confirmed by AC and noise SPICE simulations of the voltage preamplifier followed by a band-pass filter with center frequency f_op_ and variable bandwidth.

Since, in QEPAS applications, several QTFs have been employed [1,2,3,4,5,6,7,8,9,10,11,12,13,14], the same study has also been carried out for a representative QTF characterized by an f_S_ = 12.484 kHz and a quality factor of 10^4^ [9], and the same results were found. 

The reported voltage amplifier, implementing the AD8067, will be employed to validate the results obtained both with our model and with the SPICE simulation.

## Figures and Tables

**Figure 1 micromachines-14-00619-f001:**
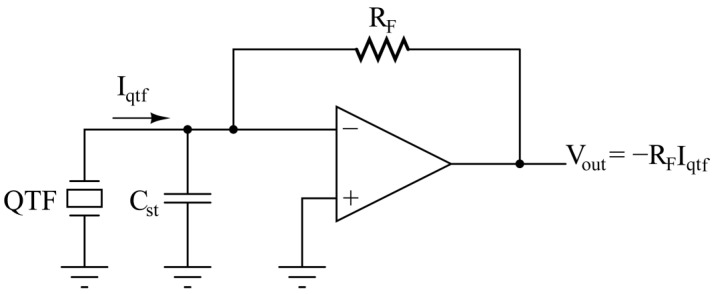
QTF read-out by means of a transimpedance preamplifier.

**Figure 2 micromachines-14-00619-f002:**
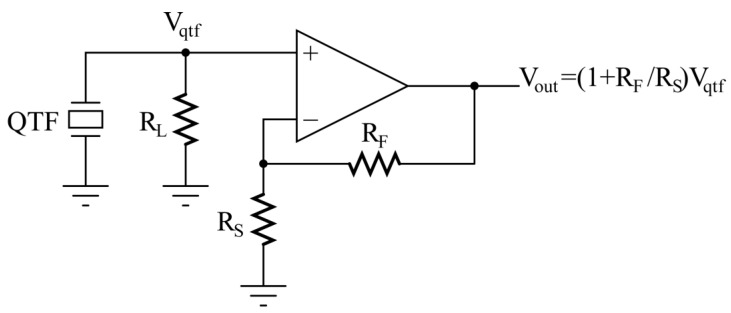
Voltage-mode read-out of the QTF (OPAMP in non-inverting configuration).

**Figure 3 micromachines-14-00619-f003:**
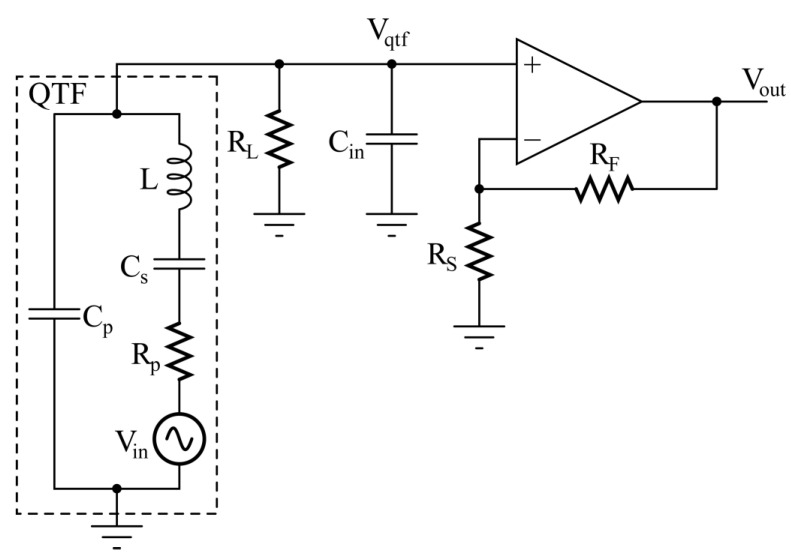
Butterworth–Van Dyke model for the QTF in the voltage-mode read-out circuit.

**Figure 4 micromachines-14-00619-f004:**
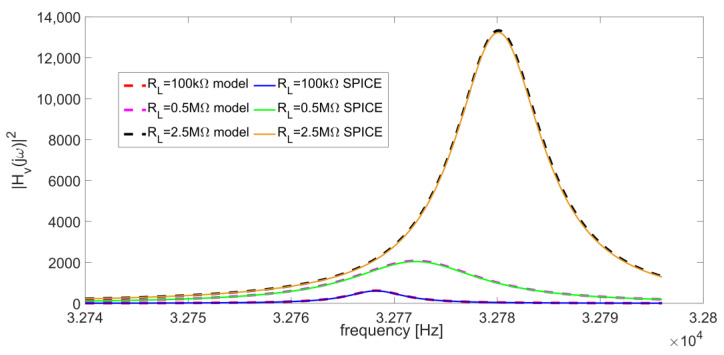
Comparison between SPICE simulations and analytical model in Equation (2) of the frequency response of the circuit in Figure 3 for R_L_ = 100 kΩ, 0.5 MΩ and 2.5 MΩ.

**Figure 5 micromachines-14-00619-f005:**
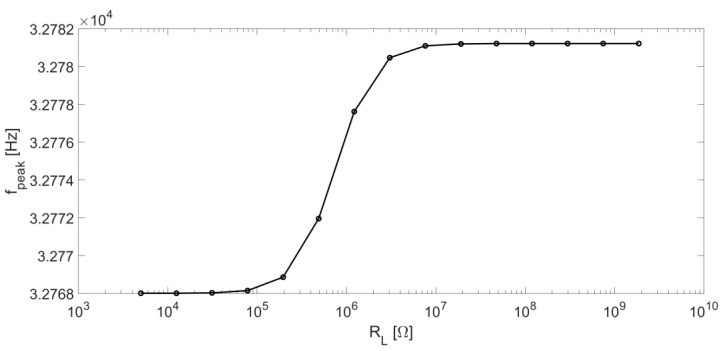
Peak frequency f_peak_ of |H_v_|^2^ as a function of R_L_.

**Figure 6 micromachines-14-00619-f006:**
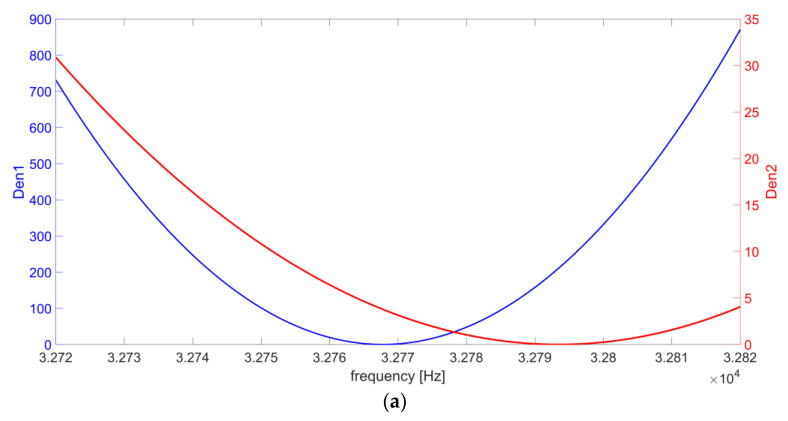
Behavior of Den1(f) and Den2(f) in Equation (3) as a function of frequency for (**a**) R_L_ = 100 kΩ and (**b**) R_L_ = 10 MΩ.

**Figure 7 micromachines-14-00619-f007:**
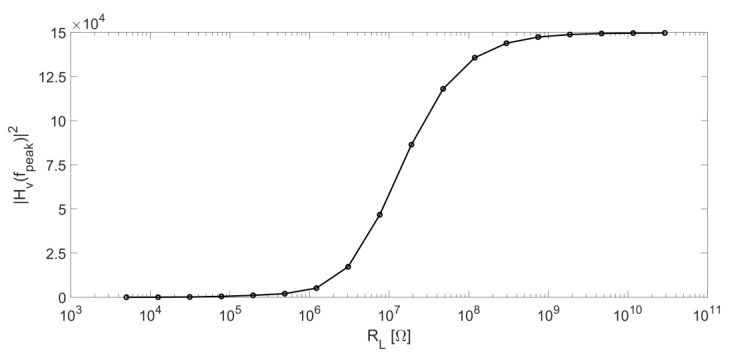
Peak value of |H_v_|^2^ as a function of R_L_.

**Figure 8 micromachines-14-00619-f008:**
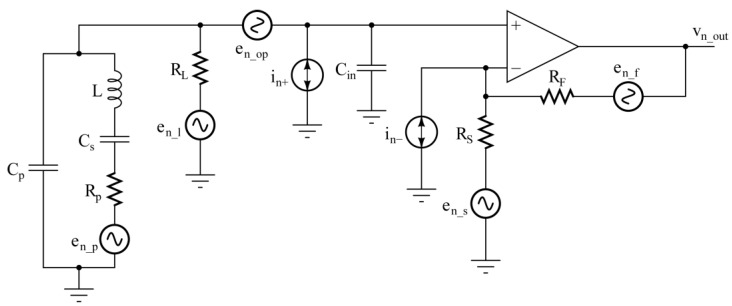
Noise contributions in the circuit of Figure 3.

**Figure 9 micromachines-14-00619-f009:**
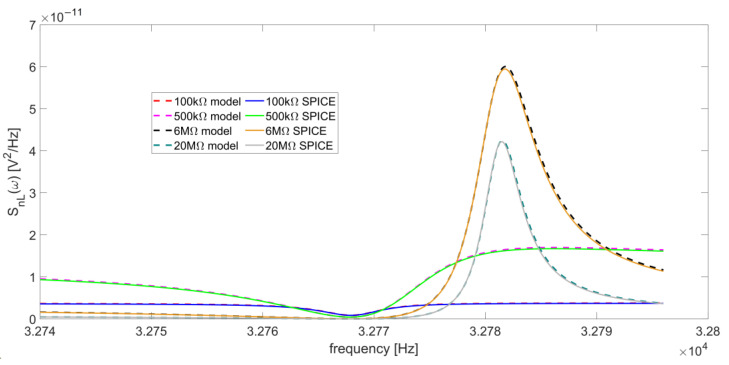
Comparison between SPICE simulations and analytical model described in Equation (7) of the output noise spectral density contribution from the thermal noise of R_L_, for four different values of the resistor: 100 kΩ, 500 kΩ, 6 MΩ, and 20 MΩ.

**Figure 10 micromachines-14-00619-f010:**
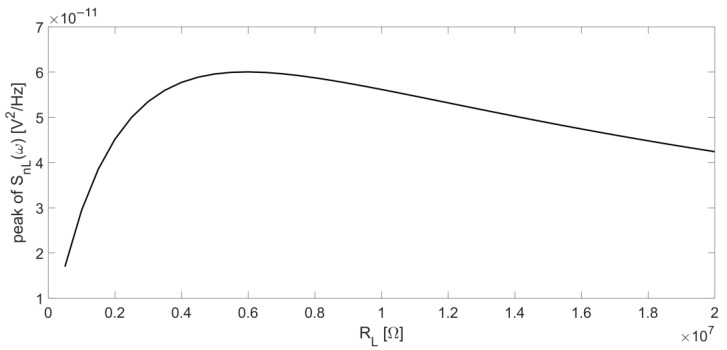
Peak of the output noise spectral density contribution due to R_L_ as a function of R_L_ itself.

**Figure 11 micromachines-14-00619-f011:**
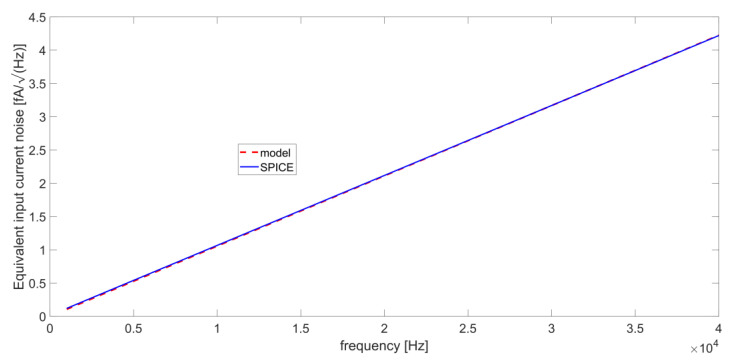
Fitting between the analytical model and SPICE simulations of the input equivalent noise current of the AD8067.

**Figure 12 micromachines-14-00619-f012:**
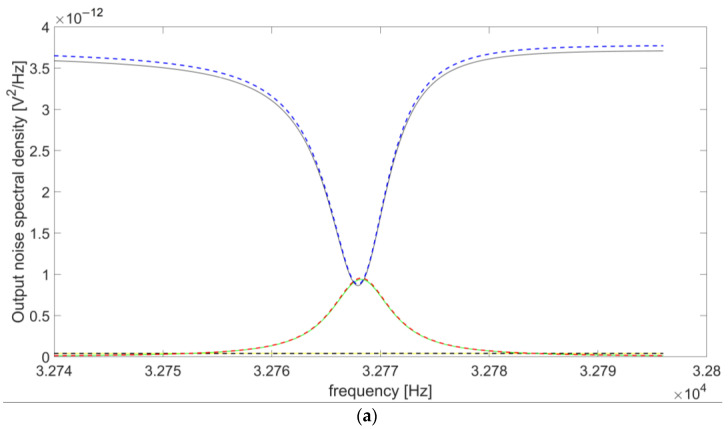
Contributions to the overall output noise spectral density due to the OPAMP and resistors R_p_ and R_L_, for (**a**) R_L_ = 100 kΩ, (**b**) R_L_ = 500 kΩ, (**c**) R_L_ = 6 MΩ, (**d**) R_L_ = 20 MΩ: comparison between the results obtained with the analytical model and SPICE simulations.

**Figure 13 micromachines-14-00619-f013:**
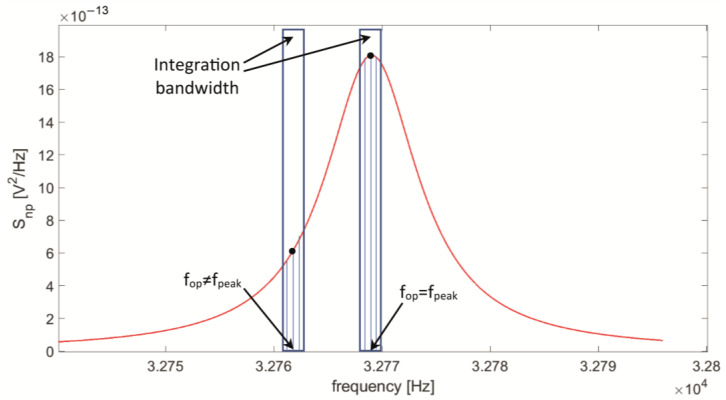
Integration of the contribution S_np_ (red line), due to R_P_, to the total output spectral noise density at a low-pass filter bandwidth of 0.5 Hz around the operating frequency f_op_, for f_op_ = f_peak_ and f_op_ ≠ f_peak_ (R_L_ = 100 kΩ).

**Figure 14 micromachines-14-00619-f014:**
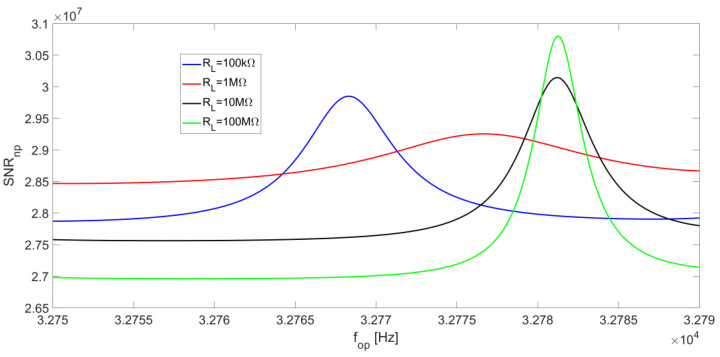
R_P_ noise contribution to the SNR_n_ at the LIA output as a function of the operating frequency, for different values of R_L_ and at a low-pass filter bandwidth of BW = 0.5 Hz.

**Figure 15 micromachines-14-00619-f015:**
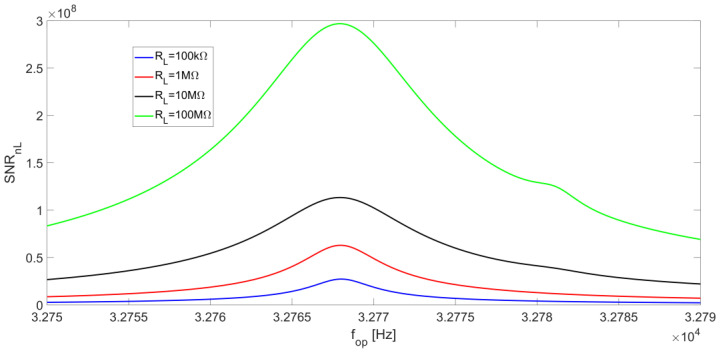
R_L_ noise contribution to the SNR_n_ at the LIA output as a function of the operating frequency, for different values of R_L_ and at a low-pass filter bandwidth of BW = 0.5 Hz.

**Figure 16 micromachines-14-00619-f016:**
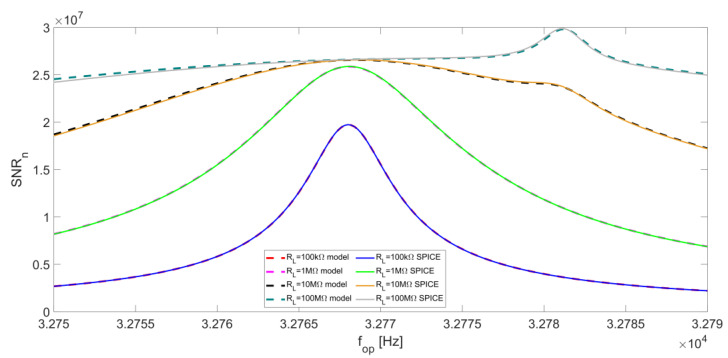
Total normalized signal-to-noise ratio SNR_n_ at the LIA output as a function of the operating frequency, for different values of R_L_ and at a low-pass filter bandwidth of BW = 0.5 Hz. The corresponding results of SPICE simulations are also reported.

**Figure 17 micromachines-14-00619-f017:**
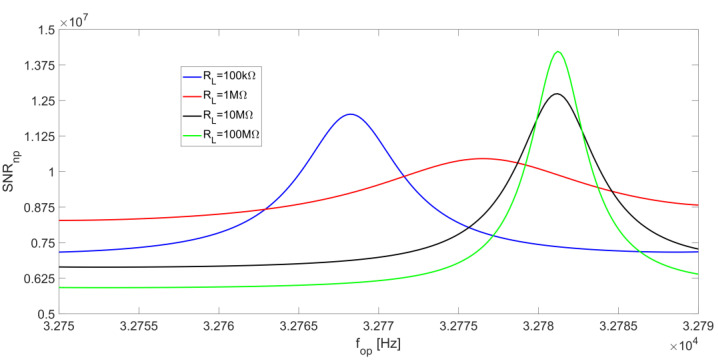
R_P_ noise contribution to the SNR_n_ at the LIA output as a function of the operating frequency, for different values of R_L_ and at a low-pass filter bandwidth of BW = 5 Hz.

**Figure 18 micromachines-14-00619-f018:**
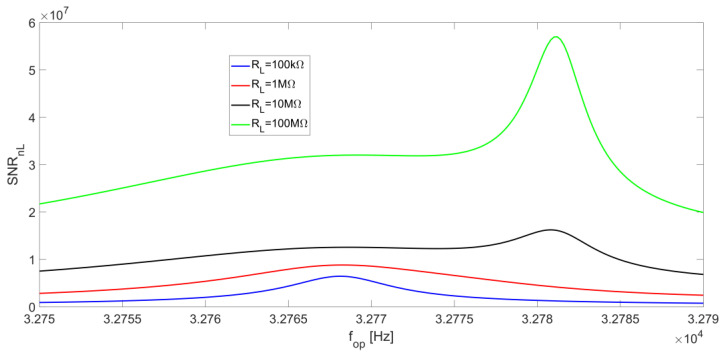
R_L_ noise contribution to the SNR_n_ at the LIA output as a function of the operating frequency, for different values of R_L_ and at a low-pass filter bandwidth of BW = 5 Hz.

**Figure 19 micromachines-14-00619-f019:**
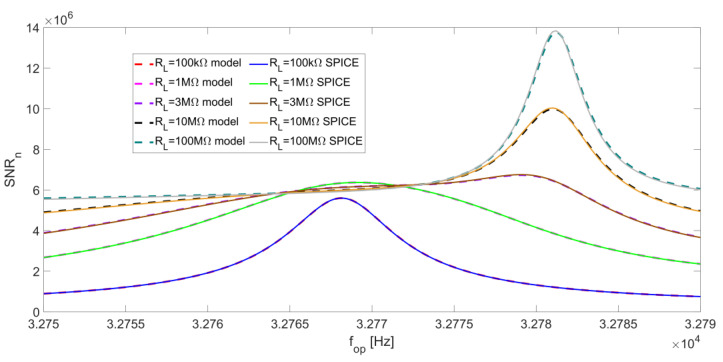
Total normalized signal-to-noise ratio SNR_n_ at the LIA output as a function of the operating frequency, for different values of R_L_ and at a low-pass filter bandwidth of BW = 5 Hz. The corresponding results of SPICE simulations are also reported.

**Table 1 micromachines-14-00619-t001:** Parameter values used to compare the results of expression in Equation (2) to SPICE simulations.

Parameter	Value
C_p_	5 pF
C_s_	5.2424 fF
L	4.5 kH
R_p_	92.7 kΩ
C_in_	1.5 pF
R_F_	47 kΩ
R_S_	1 kΩ

**Table 2 micromachines-14-00619-t002:** NNEA for different values of R_L_ and ∆f, calculated starting from the value reported in [10] for the detection of CO_2_ in an N_2_ mixture.

R_L_ [MΩ]	Δf [Hz]	NNEA [W·cm^−1^/√Hz]
10	0.5	5.0 × 10^−9^ [10]
100	0.5	4.4 × 10^−9^
10	5	5.8 × 10^−9^
100	5	4.1 × 10^−9^

## Data Availability

Not applicable.

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
