# Peer review of "Signal-to-Noise Ratio Analysis for the Voltage-Mode Read-Out of Quartz Tuning Forks in QEPAS Applications"

_micromachines, 2023, doi:10.3390/mi14030619_

Round 1
Reviewer 1 Report
The authors present the analysis of the signal to noise ratio of a conditioning circuit for Quartz Tuning Forks (QTF) consisting of a load resistor and an opamp non-inverting amplifier. They take into consideration the value of the load resistor and the equivalent bandwidth of the lock-in amplifier.
To study and compare the sources of noise they use the Butterworth-Van Dyke model of the QTF. The parameters of the model they use are ok but I think they should be better justified either by references or, if possible, by real measurements and parameter extraction.
My main concern about the paper, and it is an issue that should be revised, is that the equivalent circuit is an electrical representation of the QTF mechanical and electrical behavior and it does not represent actual circuit components. Their analytical model and I guess the SPICE simulations use circuit components, including the resistor Rp, which is part of the model but not a real resistor. If Rp is not a real resistor, then the noise figure they use don`t apply. Further investigation is required to verify that their noise model is correct.
The paper only present simulations, some experimental results would much further increase the interest of the paper.
Other comments:
- - Figure 5 and figure 7 could have the same range of RL values.
- - Figure 12 could be represented in a more compact way and using the same resistor values as in figure 9.
- - It is quite confusing to include every time the analytical curve and the SPICE simulation results.
- - They justify the QTF read-out circuit for QEPAS. At the end they show values of SNR in the order of 10 to the 7 but the numbers have no physical meaning. Authors should find a way to relate the numbers with NNEA or any other figure of merit for QEPAS.
- - Figures appear a little blurry.
Reviewer 2 Report
Journal: Micromachines
Manuscript ID: micromachines-2236743
Type of manuscript: Article
Title: Signal-to-Noise Ratio Analysis for the Voltage-Mode Read-out of Quartz Tuning Forks in QEPAS Applications
Authors: Michele Di Gioia, Luigi Lombardi, Cristoforo Marzocca, Gianvito Matarrese, Giansergio Menduni, Pietro Patimisco, Vincenzo Spagnolo
The authors reported a method via theoretical analysis of the parameters which are used to read out the signal from Quartz tuning forks (QTFs). Through the simulation of signal to noise ratio (SNR) dependence on QTF equivalent resistor and the input bias resistor RL of the preamplifier, the operating frequency (fP), and the bandwidth (BW) of the lock-in amplifier low-pass filter, they provided an effective pathway to select the better RL, fp and BW, so as to get the best SNR for better sensitivity in QEPAS applications.
I recommend this manuscript to be published on Micromachines after minor revision. I have several comments below:
1. It is a fact that QTF with different resonant frequency (f) from several kHz to several-ten kHz have been applied to QEPAS, the commercial and typical one is f=32.768 kHz. Here the authors only analyze QTF with f=32.768 kHz, whether your method can be applied to all QTFs, please give some description.
2. There are a lot of experimental results in the published papers about QTF in QEPAS, could make some intercomparisons of your model with the experiment.
3. The time response of QTF is an important factor in QEPAS for temporal-sensitive measurement, could you give some analysis of this topic using your model?
4. Based on your theoretical model, could you quantify the SNR improvement factors if the best parameters were used? The estimation is also very prospective for QEPAS application.
5. What is SPICE simulation, could you give the references or the relevant definition?

Reviewer 3 Report
Quartz-enhanced photoacoustic spectroscopy (QEPAS) is an important optomechanical sensing technique and a thorough analysis of its signal-to-noise ratio (SNR) in the voltage-mode readout scheme is surely of interest to many in the sensors community. Part of the conclusion of this work may seem to be intuitive and not very stunning for some (e.g., a large R_L usually leads to higher SNR and for a narrow LIA bandwidth and a large R_L, SNR is not very sensitive to the choice of the optimal frequency), however, I don’t think that this fact leads to any reduction in the value of the present work. For me (and probably many readers of this journal), it is very important that the QEPAS signal readout circuit can be carefully analysed and the conclusion is documented in the scientific literature. Nonetheless, before the manuscript can be accepted for publication, I would like the authors to carry out some revisions taking the following comments into consideration.
My main concern is that in a high-Q mechanical oscillator like the quartz tuning fork, the readout noise (as well as the SNR) is often heavily affected by the phase noise rather than the electronic noise alone (phase shift being converted to an amplitude fluctuation). While I perfectly understand that analysing the phase noise in QEPAS is out of the scope of this work, I still believe that the authors ought to consider adding at least one paragraph to discuss the effect of the phase noise in QEPAS detection and how it may complicate the issue being discussed in this work. Such a discussion allows a full picture to be presented to the readers, so one does not form an overtly simplified idea of the problem.
Some other minor issues are given below.
· Lines 116 – 117. This is where the AD8067 op-amp is first mentioned in the manuscript. The authors ought to add at least one sentence to provide some background information. Is this op-amp used in the QEPAS readout circuit the authors built in their lab? Why did they select this op-amp? How much does the op-amp selection affect the results presented in this study?
· Table 1 on page 4. Why do the values contain very different numbers of significant figures? How many significant figures should these values contain?
· In my opinion, it is desirable if yellow is not used to colour any curve in the figures. Its visibility is very low. This is a common issue in nearly every figure in this manuscript.
· Fig. 5 on page 5 and Fig. 7 on page 7. The authors ought to consider adding a sentence to explain how the “peak frequency” and “peak value” are found in this work. Are they defined through calculating the second derivative of the analytical expression, a curve fitting to the SPICE simulation result, etc?
· Lines 165 – 166. I don’t fully agree with the authors’ assessment here. C_S << C_pt only ensures that C_S / (C_pt + C_S) is a small number. As Table 1 indicates, C_pt / C_S ~ 1000 and R_P is of the same order of magnitude as R_L for R_L = 100 kohm. Hence, the expression (R_P / R_L) (C_S / (C_pt + C_S)) is in the order of 1/1000. However, this value has to be compared to (1 - \omega^2 / \omega_P^2)^2 in the region where \omega is close to \omega_P as shown in Fig. 6. Taking the lower limit of the horizontal x axis in Fig. 6 (f = 32720 Hz) and f_P = 32781 Hz, (1 - \omega^2 / \omega_P^2)^2 ~ 4e-3 which is of the same order of magnitude and this value becomes even smaller when \omega is approaching \omega_P. Hence, C_S << C_pt cannot make sure that the expression under discussion is negligible, at least not for R_L = 100 kohm.
· Fig. 12 on pages 12 and 13. The contrast between the curves in the figure is low. It is not easy to tell the difference between the yellow and the green curves (especially after dashed curves have been overlaid on top of them).
· The unnumbered equation on page 13. Shouldn’t this equation be numbered? Throughout the whole manuscript, the angular frequency \omega is being used (instead of the frequency f). However, suddenly starting from here and also in several other equations on page 14, the frequency f becomes the dependent variable. May it be better to make the expressions more consistent? Moreover, there should probably be a ‘j’ in the numerator of this equation.
· I find the discussion on page 14 confusing though a careful reading suggests that there is nothing fundamentally wrong in it. The authors should consider rewriting part of it to clarify a few issues. First of all, the LIA is modelled as a bi-quadratic band-pass filter (strictly speaking, there are some intricate differences between an LIA and a band-pass filter due to the phase-sensitive nature of the former, however, I agree that modelling the LIA as a bi-quadratic band-pass filter here serves the purpose of this work well), which is a second-order filter. It does not make sense to talk about a "first-order LIA low-pass filter" in line 332. Second, the usage of "integration bandwidth" in Fig. 13 is misleading. Since the LIA response is modelled as a bi-quadratic filter, the window shape cannot be rectangular as Fig. 13 seems to suggest. Third, I would argue that the most common way of defining the SNR is the ratio of the signal power to the noise power. While it is not wrong to use the voltage ratio as the authors do, it is simply not what most researchers would do.
Round 2
Reviewer 1 Report
This reviewer concerns have been clarified so my recommendation is to accept the manuscript in the present form.
Author Response
We would like to thank once again the Reviewer for his comments and observations and for his interest in our work.
Reviewer 3 Report
I thank the authors for taking all my comments into account in the revised manuscript. While I believe that the authors have addressed most of my comments very well, I have some doubts on the authors’ reply to my last comment. It is very possible that I may have misunderstood certain statements the authors made. Hence, if this is indeed the case, the authors ought to feel free to point it out to me.
First of all, the settling time of a filter depends on its roll-off. In the first version of the manuscript, line 332 – 333 reads “First, let us consider a very narrow-bandwidth BW = 0.5 Hz, corresponding to a settling time of about 1.3 s for a first order LIA low-pass filter”. I pointed out in my comments that this statement was inconsistent with the assumption that the LIA was modelled as a biquadratic (second-order) band-pass filter. In the revised manuscript, the reference to a “first order filter” has been removed (lines 350 – 351). However, as far as I can see, the settling time of 1.3 s stated here is only correct for a first-order filter. For a second-order filter, the setting time would be significantly longer.
Second, I’m not fully convinced that the equivalent noise bandwidth (ENBW) can simply be used in place of the integration bandwidth, as Fig. 13 and the revised text suggest. The definition of the ENBW assumes white noise. When the noise is non-white, e.g., S_np in the present work, the filter window shape has to be accounted for if quantitative results are desired.
I would appreciate it if the authors could take a look at the above two (minor) issues and address them as they see appropriate.
